# Manipulation Inversion by Adversarial Learning on Latent Statistical Manifold

## Abstract

The inversion of generative adversarial network (GAN) is able to investigate rich semantics within the generative models, thus receiving increasing research efforts most recently. Existing GAN inversion methods focus on reconstructing images, with relatively less focus on improving the editing realism, the most important criterion for evaluating the semantics achieved by inversion. In this paper, we systematically investigate the latent generating space and prove that both the realism of editing and accuracy of reconstruction can be unified under the umbrella of the inversion against manipulations. Motivated by this, we propose to establish the generating space as latent probabilistic models, followed by the developed statistical manifold to minimise the distribution discrepancy. Based on the manifold, we further propose an adversarial learning strategy to avoid the excessive enumeration when calculating the manipulation inversion metric. We may also need to point out that the proposed method is universal to different architectures, as a novel plugin inversion method. We comprehensively evaluate our method across different types of network architectures, comparing it against the state-of-the-art inversion methods. The experimental results demonstrate that our method is able to achieve superior performances on both reconstruction accuracy and realism of editing.

## 1. Introduction

Generative adversarial networks (GANs) have been playing as the cutting-edge deep generative models for generating realistic content (Sauer et al., 2022; Kang et al., 2023), which also popularises its application to various tasks such as image/video compression (Mentzer et al., 2020; 2022), super-resolution (Wang et al., 2021; 2018), enhancement (Galteri et al., 2019), to name but a few. Compared with existing deep generative models, the merit of GANs arises from the distinct intrinsic nature of directly generating highly realistic images from low-dimensional random noise, thus capable of depicting the complicated high-dimensional data from the low-dimensional continuous latent generating space. This merit also enables the latent generating space to possess rich and precise semantics (Shen et al., 2020b; Härkönen et al., 2020), as the potentially well-behaved proxy for representing the real-world scenarios.

Since almost all the existing GANs uni-directionally generate images from the latent space, the way to invert images back into the latent generating space of GANs is the prerequisite before we start to investigate the rich semantics from real-world scenarios. This essentially requires carefully embedding into the latent space to ensure both the accuracy of reconstruction and realism of editing, which are oftentimes trade-off with the other (Wang et al., 2022; Dinh et al., 2022; Tov et al., 2021; Yao et al., 2022). Existing methods address this trade-off by restoring the semantics in the latent space and reconstructing the details in the middle-layer features, given a pre-trained (or slightly fine-tuned) generator (Wang et al., 2022; Dinh et al., 2022; Yao et al., 2022). However, existing GAN inversion methods essentially focus on point-wise estimation against the latent representation for both inverting images and semantics, without considering the characteristics of arising local curvature, thus suffering from the incompleteness regarding editing based on the estimated point. Thus, the improvement on the editing performance, the key to depicting the latent space characteristics, is still of an *ad hoc* manner (Xia et al., 2022).

Indeed, the preferred GAN inversion is able to embed arbitrary realistic images into the latent space, followed by accurate reconstruction based on the embedding; this is expected to still hold when embedding and restoring edited images, given the fact that the realism of editing is another criteria of inverting GANs. Therefore, the desirable embedding is capable of accurately inverting both the original image and its arbitrarily edited counterparts, which as shall be proved in this paper, is equivalent to the capability of precisely restoring the edited image back to the original image; we name this operation as *inverting manipulations* that essentially poses more stringent requirements against inverting images and semantics of GANs. Unfortunately, although the concept has been preliminarily mentioned in a few works, by either evaluation metrics (Tov et al., 2021) or auxiliary cycle consistency regularisation (Pehlivan et al., 2023), inverting manipulations is still yet to be systematically investigated by far.

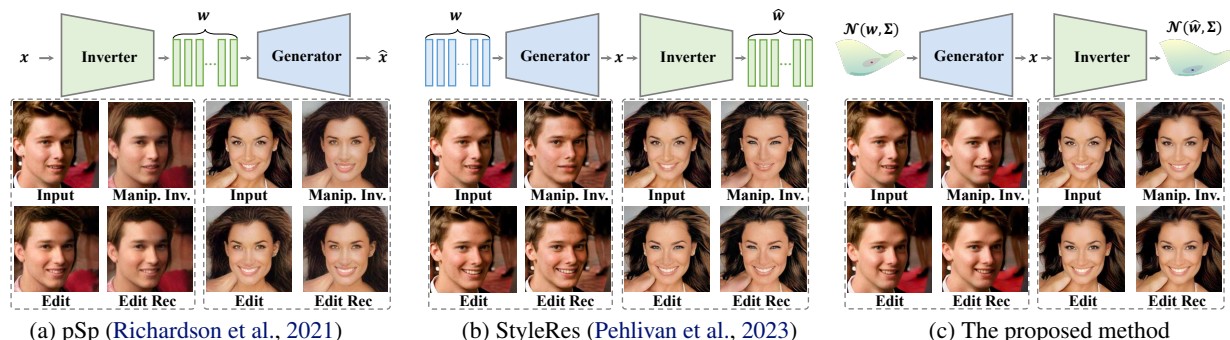

(a) pSp (Richardson et al., 2021)    (b) StyleRes (Pehlivan et al., 2023)    (c) The proposed method

*Figure 1.* Illustration of our method and existing typical inversion methods. The pixel2Style2pixel (pSp) method focuses solely on the image-domain reconstruction, while StyleRes method imposes additional regularisations in the latent space. Existing methods essentially optimise point-wise error and thus fail to consider local curvature in the latent space, leading to the inaccurate reconstruction of edited images and manipulation inversion. In contrast, our method optimises the inversion of manipulation based on establishing a statistical manifold in the latent space, which is able to achieve superior performances on both reconstruction accuracy and editing realism. Please note that *Manip. inv.* denotes the inversion of manipulation, while *Edit Rec.* denotes the reconstruction of the edited image.

In this paper, we set out the first attempt to invert arbitrary manipulations upon GANs, so as to optimise both the latent representation and its corresponding local curvature, as illustrated in Fig. 1. More specifically, we first systematically analyse the characteristics of GANs, including in-depth analysis on the local optimum and curvatures within the generating space. In light of the analysis, we propose to embed each inverting image into an individual distribution, in which randomly sampling from the distribution operates as variants within the same identity of images, including semantic editing and non-semantic nuisance noise to reflect the local curvature. We then establish the statistical manifold for the GAN generating space based on the Cramer-Rao metric, and optimisation on the manifold improves both the image reconstruction and manipulation inversion.

Therefore, we propose to optimise the inversion of manipulation based on the established manifold, the goal that cannot be achieved by the *de facto* point-wise reconstruction by almost all inversion methods. To further relieve the excessive enumeration of random samples for inverting arbitrary manipulations, we propose an adversarial strategy to efficiently reduce the searching trials during the optimisation procedure. This way, we are able to unify the optimisation of manipulation inversion problem, under an efficient end-to-end distribution alignment within the latent space in practice. Consequently, experimental results verify the superior performance of our method in precisely inverting the manipulation, as well as on the accuracy of reconstruction and the quality of editing.

## 2. Related Works

Since the StyleGAN architecture has been exhibiting the state-of-the-art generation performances in various scenarios, existing GAN inversion methods mainly focus on the StyleGAN architecture (Karras et al., 2019; 2020; 2021), in which the images are generated sequentially from the random noise $\mathbf{z}$, the style code $\mathbf{w}$ and the transformed style codes $\mathbf{w}^+$. We thus name the corresponding spaces as $\mathcal{Z}$, $\mathcal{W}$ and $\mathcal{W}^+$, respectively.

**Inversion on Images:** Existing methods regarding inverting StyleGANs can be generally categorised into three groups, the optimisation-oriented, encoder-based, and hybrid methods. Based on either gradient descent solvers (Yeh et al., 2017; Zhu et al., 2016; Fang & Schwing, 2019) or gradient-free strategies (Huh et al., 2020; Abdal et al., 2019; 2020), the optimisation-oriented methods exhaustively seek the best latent representation for each image , at the cost of heavy computational complexity . On the other hand, the encoder-based methods focus on achieving universal inversion, with the goal to learn general solutions regarding image inversion. The hierarchical encoder architecture is typically employed to embed multiple scales into transformed styles $\mathcal{W}^+$ (Richardson et al., 2021) . Advanced inversion methods accommodate the reconstruction-editing trade-off by a two-phase strategy, in which the first phase aims to retain the editing ability in the $\mathcal{W}$ (or $\mathcal{W}^+$) space, and additional modules are developed in the second phase so as to compensate the reconstruction error (Wang et al., 2022; Dinh et al., 2022; Li et al., 2023; Pehlivan et al., 2023) . The above encoders can also be combined with the optimisation-oriented methods, in which the encoders provide a well-defined initialisation for the optimisation-oriented methods (Zhu et al., 2016; Hussein et al., 2020; Roich et al., 2022; Alaluf et al., 2022). However, all the above methods mainly focus on inverting images, which fall short in retaining the local curvature and thus inevitably exhibit deficiency on inverting manipulations of StyleGANs.

**Inversion on Latent Representations:** Since the latent spaces of StyleGANs including $\mathcal{Z}$ and $\mathcal{W}$ spaces have been proved to possess rich semantics (Härkönen et al., 2020; Shen et al., 2020b; Abdal et al., 2019), we also witnessed several recent inversion methods that regularise the align-

ment within the latent spaces (Tov et al., 2021; Bau et al., 2019; Zhu et al., 2020; 2024). Latent space regularisation can find its roots in bi-direction generation of training GANs, by either catering for theoretical completeness (Li et al., 2022) or practice benefits (Ding et al., 2020; Dumoulin et al., 2016). However, since their primary goal focuses on the generation quality, these methods still suffer from inaccurate restoration of semantics and reconstruction of images. Regarding inversion based on pre-trained GANs, in addition to reconstructing images by pixel-wise loss, the E4E method (Tov et al., 2021) also develops a discriminator in the $\mathcal{W}$ space, so as to regularise the latent representations from the trained encoder to be similar to the original generating space, and correspondingly proposed a metric called latent editing consistency to measure the editing capability. On the other hand, Bau *et al.* (Bau et al., 2019) proposed to pre-train an encoder by inverting the randomly sampled latent representation in $\mathcal{Z}$ space, which is then used for the following layer-wise optimisation. Zhu *et al.* (Zhu et al., 2020) further proposed an optimisation-oriented method, which inverts images with the assistant of in-domain image prior in the $\mathcal{Z}$ space. However, for the encoder-based methods, it is obvious that without any additional constraints, the inversion in the latent space is ill-posed, since the minimisation of $||f(\mathbf{x}) - f(g(f(\mathbf{x})))||_2^2$ can find its bad local minimum at 0 for any surjection $f(\mathbf{x}) = c$, where $f$ is the encoder to be optimised, $\mathbf{x}$ represents the input image, $g$ denotes the fixed generator and $c$ is any constant. More importantly, existing methods in the latent space are based on the point-wise estimation. As shall be shown shortly, the point-wise estimation, oftentimes taken for granted, is proved to be insufficient in indicating the semantic discrepancy.

## 3. Analysis on Latent Generating Space

Compared with existing GAN models, the StyleGAN architecture has achieved the state-of-the-art performance when generating realistic and diversifying images (Karras et al., 2020; Sauer et al., 2022); this essentially ensures the richness of semantics in the latent generating space (Abdal et al., 2019; 2020). Therefore, we mainly focus on analysing the latent space of StyleGAN, by revealing several important findings and properties that motivate our follow-up method for inverting the manipulations.

***Finding 1:*** *There exist multiple local domains in the latent space that correspond to the same person identities.*

We first analyse the correspondence between the latent space and generated images, based on the StyleGAN model. More specifically, our analysis is based on the officially released model (Karras et al., 2020), which is adopted in almost all inversion methods based on StyleGAN. By inspecting the generator, we essentially find that the domains of $\mathbf{w} \in \mathcal{W}$ and $\mathbf{w}^+ \in \mathcal{W}^+$ obtained by sampling $\mathbf{z}$ from the standard

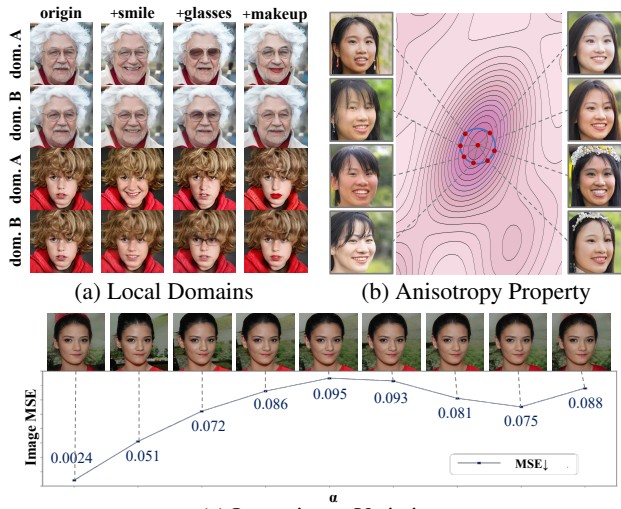

(a) Local Domains      (b) Anisotropy Property

(c) Inconsistent Variation

*Figure 2.* In-depth analysis on the lantent generating space of Style-GAN. (a) represents multiple local domains corresponding to the same person identities. (b) represents the anisotropy property across different directions. (c) denotes the inconsistency between the latent space and image space.

Gaussian distribution (named as the *sampling domain*) are not well aligned with those obtained from inverting methods (named as the *inverting domain*). The average $l_2$ norm distance between those two domains is much larger than that between the generated images from the two domains, exhibiting that those two domains are well separated in the latent space. More importantly, both of the two domains, even with their interpolations, are able to reconstruct extremely similar images with the same person identity, as illustrated in Fig. 2-(a). This reveals that the similarity of two images may not be sufficient to guarantee the closeness in the latent space. On the other hand, the similar images across the interpolation indicate that certain direction in the latent space cannot alter the image semantics, which is further analysed by the following findings.

***Finding 2:*** *When manipulated by directions with the same scale, the generated images exhibit distinct anisotropy across semantics.*

The rich semantics within the low-dimensional latent space allow for flexible manipulations. We investigate the impact of manipulations on the $\mathcal{W}$ space, the *de facto* choice for the majority GAN inversion methods (Richardson et al., 2021; Tov et al., 2021; Alaluf et al., 2022; Dinh et al., 2022; Alaluf et al., 2021; Hu et al., 2022; Pehlivan et al., 2023; Wang et al., 2022; Yao et al., 2022). We represent the generation from the $\mathcal{W}$ space as $\{g(\mathbf{w}) : \mathbf{w} \in \mathcal{W}\}$, and analyse the manipulated generation $g(\mathbf{w} + \mathbf{v})$ for arbitrary $\mathbf{v}$ that satisfies both $||\mathbf{v}||_2^2 = \beta$ and $(\mathbf{w} + \mathbf{v}) \in \mathcal{W}$. Please note that $\beta$ is the constant that restricts the scale of manipulating direction $\mathbf{v}$ to possess the same length of the $l_2$-norm. We then plot sets of $g(\mathbf{w} + \mathbf{v})$ and $g(\mathbf{w})$ for different $\mathbf{v}$ in Fig. 2-(b). From this figure, when manipulated by the

same scale vector within the $\mathcal{W}$ space, we can conclude that the variations of generated images are distinct, in which certain random directions eventually anisotropically change generated person identities. In other words, the variation of an image within the same person identity essentially corresponds to a curved latent space.

*Finding 3: When sequentially edited by a fixed semantic direction, the generated images exhibit inconsistent variation against the unedited image.*

Given the fact that GAN inversion searches for the best representation in the latent space to minimise the discrepancy between generated and input images, we then analyse the relationship between the deviation in the latent space and the variation of the corresponding generated images. More specifically, we investigate the deviation on the $\mathcal{W}$ space by directions with explicit semantics, which can be obtained by InterfaceGAN (Shen et al., 2020a) for the face images and GANSpace (Härkönen et al., 2020) for the car and church images. Given a normalised semantic direction $\{\mathbf{e} : ||\mathbf{e}||_2 = 1\}$, we are able to calculate the variation of generated images by $||g(\mathbf{w}) - g(\mathbf{w} + \alpha \cdot \mathbf{e})||_2^2$, in which $\mathbf{w} \in \mathcal{W}$, $\alpha \in \mathbb{R}^1$ denotes the deviation scale, $g(\cdot)$ denotes the generation process and the variation is evaluated by the MSE metric $|| \cdot ||_2^2$. We illustrate in Fig. 2-(c) regarding the MSE values between edited and unedited images, along with the change of scale $\alpha$ in the latent space. From this figure, we can conclude that when increasing the scales given a direction, the generated images, although still possessing the same identity, exhibit inconsistent MSE values, sometimes even have decreased MSE results. In other words, given fixed (or slightly fine-tuned) generators, minimising MSE on images may even result into the increase of deviation in the latent space, thus preventing from finding the best latent representation. In contrast, considering the curvature within the latent space is beneficial to achieve the global optima.

## 4. Methodology

### 4.1. Latent Manipulation Inversion

Basically, GAN inversion seeks to accurately restore realistic images, whereas *Finding 1* indicates that directly inverting images can result into sub-optimal results due to multiple local domains in the latent space. Since the latent space of StyleGAN possesses rich semantics, we propose to restrict the inversion within the latent space to ensure the consistency on semantics. Indeed, the inversion essentially requires the bijection from the encoder to the generator for real-world images, i.e., $\mathbf{x} = g(f(\mathbf{x}))$, which is equivalent to the bijection from the generator to the encoder within a certain local domain, namely, $\mathbf{w} = f(g(\mathbf{w}))$. Therefore, performing the inversion within the latent space can also contribute to improving the inversion for restored images.

The other important criteria of inversion is the quality of semantics of the inverted latent feature, i.e., retaining the realism of editing. Given the fact that the GAN inversion restores realistic images, the preferred GAN inversion thus has to restore arbitrarily edited images. When **Assumption 1** exists, **Lemma** 4.2 ensures the equivalence between inverting arbitrarily edited images and inverting arbitrary manipulation in the latent space, thus providing a new way of improving the GAN inversion.

**Assumption 4.1.** The generator of StyleGAN is locally Lipschitz and operates as continuous mapping from the latent space $\mathcal{W}$ to the image space (Arjovsky et al., 2017). More importantly, there exists a local domain that the generation is injective, which is also the prerequisite for GAN inversion.

**Lemma 4.2.** *Let $g(\cdot)$ denote the pre-trained generator, and $f(\cdot)$ to represent the inversion encoder. Given an arbitrary latent feature $\mathbf{w}$ from an image $\mathbf{w} = f(\mathbf{x})$ and direction $\mathbf{v} \in \mathcal{B}_\epsilon(\mathbf{w})$, where $\mathcal{B}_\epsilon(\mathbf{w})$ represents an open ball of $\mathbf{w}$ with radius $\epsilon$, we represent the edited image by $\widetilde{\mathbf{x}} = g(\mathbf{w} + \mathbf{v})$. Then, the arbitrarily edited image $\widetilde{\mathbf{x}}$ can be precisely inverted, i.e., $\widetilde{\mathbf{x}} = g(f(\widetilde{\mathbf{x}}))$, if and only if we are able to invert the manipulation, i.e., $f(\widetilde{\mathbf{x}}) - \mathbf{v} = f(\mathbf{x})$.*

*Proof.* Please refer to the Appendix-C for the proof. □

Besides performing the inversion for the latent features, we thus propose to invert the manipulation within the latent space, which basically calls for the consistency of local curvature surrounding the inverted latent feature. Given a style code $\mathbf{w}$, manipulation inversion can be formally achieved by minimising the follow objective:

$$\min_f \mathcal{L}_r = \min_f \int ||f(g(\mathbf{w} + \beta \frac{\mathbf{v}}{||\mathbf{v}||_2})) - \beta \frac{\mathbf{v}}{||\mathbf{v}||_2} - \mathbf{w}||_2^2 d\mathbf{v}, \tag{1}$$

where $\mathbf{v}/||\mathbf{v}||_2$ denotes unit manipulation and $\beta$ denotes the constant scale to retain within the same identity. In (1), recall that $g(\cdot)$ denotes the fixed generator and $f(\cdot)$ represents the inversion encoder to be optimised. As proved by **Lemma** 4.2, minimising (1) essentially ensures the ability of restoring arbitrarily edited images.

### 4.2. Latent Statistical Manifold

We propose to align the distributions within the latent space, named as distribution preserving embedding (DPE), as a well-defined proxy of local curvature. This essentially requires to establish the latent probabilistic model for $\mathbf{w}$. As analysed by *Finding 2*, the latent style features $\mathbf{w}$ exhibit anisotropic property across image semantics, and we thus cannot rely on the isotropic Gaussian assumption that is employed in typical settings. Correspondingly, we extend the Gaussian model in (Wulff & Torralba, 2020) to the widely applied factor model for latent codes, as follows,

$$\mathbf{w} = \mathbf{S}^T \mathbf{n} + \boldsymbol{\epsilon} + \mathbf{c}, \tag{2}$$

where $\mathbf{c}$ relates to the conditions given by the inversion encoder $f(\mathbf{x})$ and mapping network $h(\mathbf{z})$, $\mathbf{S}$ is the projection matrix, $\mathbf{n}$ and $\epsilon$ denote two independent random variables that satisfy Gaussian distributions. Please note that $h(\mathbf{z})$ is fixed and $f(\mathbf{x})$ is encouraged to approach $h(\mathbf{z})$, such that the manipulation inversion in the latent style code is optimised.

More importantly, since the style code feature $\mathbf{w} \in \mathcal{W}$ possesses rich semantics as reflected by *Findings 1&2&3*, we choose $\mathbf{S}$ as the semantic matrix, in which each column of $\mathbf{S}$ represents one semantic direction. As proved in various works (Shen et al., 2020b), all the latent vectors corresponding to the same attribute of generated images should be reachable through the direct path between them, and the direct path is chosen as one semantic direction in $\mathbf{S}$. In this way, $\mathbf{n} \sim \mathcal{N}(\mathbf{0}, \mathbf{I})$ and $\mathbf{S}^T\mathbf{n}$ randomly combines various semantic directions to generate the style code, which also facilitates diverse and complete semantics of generated images. On the other hand, $\epsilon$ essentially represents the nuisance noise that denotes the randomness of generated images whilst not altering the semantics. This can be established by $\epsilon = \mathbf{J}^T\boldsymbol{\eta}$, whereby $\boldsymbol{\eta} \sim \mathcal{N}(\mathbf{0}, \mathbf{I})$ denotes the random Gaussian noise on the image, and $\mathbf{J}$ is the Jacobian matrix when generating images from $\mathbf{w} \in \mathcal{W}$, which maps the nuisance noise at the image side to latent style codes.

Therefore, according to (2), we are able to model the distributions output from the inversion encoder $f(\mathbf{x})$ and mapping network $h(\mathbf{z})$, as $\mathcal{N}(f(\mathbf{x}), \mathbf{S}^T\mathbf{S} + \mathbf{J}^T\mathbf{J})$ and $\mathcal{N}(h(\mathbf{z}), \mathbf{S}^T\mathbf{S} + \mathbf{J}^T\mathbf{J})$, respectively. This way, we can minimise the distance between the two Gaussian distributions, so as to accommodate the manipulation inversion in (1) by random directions $\mathbf{v}/||\mathbf{v}||$. More importantly, $f(\mathbf{x})$ and $h(\mathbf{z})$ now represent two distributions. In other words, given two Gaussian distributions, the way to optimise $f(\mathbf{x})$ shall follow the shortest path given by the distribution discrepancy between $\mathcal{N}(f(\mathbf{x}), \mathbf{S}^T\mathbf{S} + \mathbf{J}^T\mathbf{J})$ and $\mathcal{N}(h(\mathbf{z}), \mathbf{S}^T\mathbf{S} + \mathbf{J}^T\mathbf{J})$. This naturally motivates us to establish the statistical manifold $\mathcal{M}_{\mathbf{w}}$ for two Gaussian distributions, by the Cramer-Rao distance (Amari, 2016), in which the Riemannian metric is

$$ds^2 = d\mathbf{w}^T(\mathbf{S}^T\mathbf{S} + \mathbf{J}^T\mathbf{J})^{-1}d\mathbf{w} \quad (3)$$

More importantly, taking advantages of the equivalence against the inner product of directional derivative on the Riemannian manifold and within the Euclidean space (Absil et al., 2008), we are able to calculate the Riemannian gradient on the established statistical manifold $\mathcal{M}_{\mathbf{w}}$. More specifically, given any smooth loss function $\phi(\mathbf{w})$ and any directional derivative $d\boldsymbol{\xi}$, we can calculate the gradient on the Riemannian manifold, i.e., Riemannian manifold, by the following equivalence

$$\nabla_r\phi(\mathbf{w})^T(\mathbf{S}^T\mathbf{S} + \mathbf{J}^T\mathbf{J})^{-1}d\boldsymbol{\xi} = \nabla_e\phi(\mathbf{w})^Td\boldsymbol{\xi}, \quad (4)$$

where $\nabla_r\phi(\mathbf{w})$ denotes the Riemannian gradient on $\mathcal{M}_{\mathbf{w}}$ and $\nabla_e\phi(\mathbf{w})$ is the Euclidean gradient. As (4) holds for

arbitrary directional derivative, we can choose linear independent directional derivatives $d\boldsymbol{\xi}$ to compose a full-rank matrix $\boldsymbol{\Lambda}$. Then, we have

$$\nabla_r\phi(\mathbf{w})^T(\mathbf{S}^T\mathbf{S} + \mathbf{J}^T\mathbf{J})^{-1}\boldsymbol{\Lambda} = \nabla_r\phi(\mathbf{w})^T\boldsymbol{\Lambda}, \quad (5)$$

such that the Riemannian gradient is obtained by

$$\nabla_r\phi(\mathbf{w}) = \nabla_e\phi(\mathbf{w})^T(\mathbf{S}^T\mathbf{S} + \mathbf{J}^T\mathbf{J}). \quad (6)$$

In practice, we follow (Shen et al., 2020b) to calculate the semantic matrix $\mathbf{S}$ and (Ramesh et al., 2018) to calculate the Jacobian matrix $\mathbf{J}$.

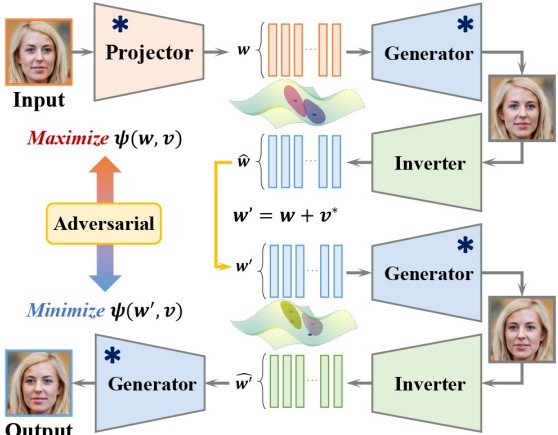

*Figure 3.* The pipeline of the proposed method. The projector first embeds existing images into a well-behaved local domain, in which the manipulation inversion is optimised based on (8). Please note that the inverter is to be optimised, whereas the generator and projector are fixed.

### 4.3. Adversarial Learning to Invert Manipulation

The remaining task is to invert the manipulation by minimising (1), based on the established manifold $\mathcal{M}_{\mathbf{w}}$. More importantly, the inversion of manipulation requires the excessive numeration on the manipulation direction $\mathbf{v}$ to calculate the integration. This, however, is intractable in practice. Although matching the distributions in the latent style code space can contribute to the manipulation inversion, an accurate inversion still requires to enumerate $\mathbf{v}$ based on (1). To relieve this issue, we propose to use the adversarial learning to choose the "best" direction that maximises (1), namely,

$$\mathcal{L}_r^* = \max_{\mathbf{v}} ||f(g(\mathbf{w} + \beta\frac{\mathbf{v}}{||\mathbf{v}||_2})) - \beta\frac{\mathbf{v}}{||\mathbf{v}||_2} - \mathbf{w}||_2^2. \quad (7)$$

The adversarial learning $\mathcal{L}_r^*$ essentially operates as an upper bound of (1). This way, minimising $\mathcal{L}_r^*$ can ensure the minimisation of (1) to retain the manipulation inversion.

To achieve the adversarial learning, we formulate the properties of the loss function $\psi$, which is represented as:

$$\psi(\mathbf{w}, \mathbf{v}) = ||f(g(\mathbf{w} + \beta\frac{\mathbf{v}}{||\mathbf{v}||_2})) - \beta\frac{\mathbf{v}}{||\mathbf{v}||_2} - \mathbf{w}||_2^2 \quad (8)$$

*Table 1.* Evaluation against the manipulation inversion among ours and existing state-of-the-art methods, on the human face, church and car scenarios. The best performance is highlighted in *red* and the second-best performance in *blue*.

| Method | Human Face | | | | Church | | | | Cars | | | |
|---|---|---|---|---|---|---|---|---|---|---|---|---|
| | MSE ↓ | LPIPS ↓ | SSIM ↑ | MS-SSIM ↑ | MSE ↓ | LPIPS ↓ | SSIM ↑ | MS-SSIM ↑ | MSE ↓ | LPIPS ↓ | SSIM ↑ | MS-SSIM ↑ |
| pSp (Richardson et al., 2021) | 0.0500 | 0.3656 | 0.5875 | 0.7196 | 0.1103 | 0.8279 | 0.3879 | 0.4226 | 0.7339 | 1.6028 | 0.2148 | 0.0564 |
| E4E (Tov et al., 2021) | 0.0665 | 0.4299 | 0.5668 | 0.6795 | 0.1713 | 1.0159 | 0.3604 | 0.3085 | 0.5290 | 1.2436 | 0.2394 | 0.0591 |
| ReStyle$_{pSp}$ (Alaluf et al., 2021) | 0.0402 | 0.2701 | 0.6057 | 0.7608 | 0.1822 | 0.7230 | 0.3589 | 0.4177 | 0.1493 | 0.6181 | 0.4922 | 0.5770 |
| ReStyle$_{E4E}$ (Alaluf et al., 2021) | 0.0602 | 0.3961 | 0.5698 | 0.7022 | 0.2593 | 1.0308 | 0.3053 | 0.2636 | 0.2922 | 0.8751 | 0.4372 | 0.4539 |
| HyperInverter (Dinh et al., 2022) | 0.0262 | *0.1645* | 0.6594 | 0.8190 | 0.0921 | 0.3815 | 0.4248 | 0.6034 | - | - | - | - |
| HFGI (Wang et al., 2022) | 0.0446 | 0.3198 | 0.5817 | 0.7481 | 0.1566 | 0.9032 | 0.3642 | 0.3811 | - | - | - | - |
| FSE (Yao et al., 2022) | *0.0223* | 0.1839 | *0.7115* | *0.8625* | 0.0573 | 0.3275 | 0.4883 | 0.7236 | *0.0772* | *0.3617* | *0.5399* | *0.7092* |
| E2Style (Wei et al., 2022) | 0.0481 | 0.4148 | 0.6253 | 0.7590 | *0.0554* | *0.3097* | *0.5244* | *0.7538* | - | - | - | - |
| StyleRes (Pehlivan et al., 2023) | 0.0366 | 0.5707 | 0.6440 | 0.7205 | - | - | - | - | - | - | - | - |
| **Ours** | *0.0139* | *0.1263* | *0.7414* | *0.8931* | *0.0458* | *0.2691* | *0.5437* | *0.7847* | *0.0486* | *0.3001* | *0.5987* | *0.7726* |

Then, we define the distance $\mathcal{D}(\mathbf{w}, \mathbf{v}) = ||\psi(\mathbf{w}, \mathbf{v}) - \psi(\mathbf{w}, \mathbf{0})||_2^2$, and approximate this using a Taylor expansion. The virtual editing reaches a maximum $\mathbf{v}^*$ through the power iteration method applied to the principal eigenvector of the Hessian (Golub & der Vorst, 2000).

In practice, to optimise the manipulation inversion within a well-behaved latent space, we employ a fixed encoder as the projector to generate $\mathbf{w}$, as illustrated in Fig. 3. We then calculate $\psi$ and update $\mathbf{w}$. After obtaining $\mathbf{w}' = \mathbf{w} + \mathbf{v}^*$, we optimise the inverter by minimizing the objective in (1), ultimately generating the final images. Therefore, our final loss becomes

$$\mathcal{L} = \mathcal{L}_r + \lambda_1 \mathcal{L}_j + \lambda_2 \mathcal{L}_s + \lambda_3 \mathcal{L}_{\text{ori}} \qquad (9)$$

where $\lambda_1$, $\lambda_2$ and $\lambda_3$ are hyperparameters and are empirically set to 1.0, 0.8 and 3.0, respectively, while $\mathcal{L}_{\text{ori}}$ represents the original inversion loss, i.e., $\mathcal{L}_{\text{ori}} = \mathcal{L}_{\text{mse}} + \lambda_{o1} \mathcal{L}_{\text{lpips}} + \lambda_{o2} \mathcal{L}_{\text{id}}$.

## 5. Experiment

### 5.1. Experimental Settings

**Dataset:** Our experimental evaluations were performed based on various scenarios. For the widely tested human face scenarios, we employed the high-quality face dataset, i.e., Flickr-Faces-HQ Dataset (FFHQ) (Karras et al., 2019) dataset for training, and evaluated based on the CelebA-HQ (Karras et al., 2018) dataset for the inversion. Both resolution for the FFHQ and CelebA-HQ datasets is $1024 \times 1024$. We also evaluated our method for the car scenario, based on the $512 \times 512$ images within the Stanford Cars (Krause et al., 2013) to serve for training and evaluation with the official split. We further evaluated the challenging scenery images by the church scenario, including $256 \times 256$ images within the LSUN Church dataset (Yu et al., 2015) and also followed the official data splitting strategy.

**Baselines** We compared our method with state-of-the-art image inversion methods, including classical methods such as pixel2style2pixel (pSp) (Richardson et al., 2021) and encoder for editing (E4E) (Tov et al., 2021), and most recent methods such as residual-based StyleGAN (ReStyle) (Alaluf et al., 2021) and E2Style (Wei et al., 2022), HFGI (Wang et al., 2022), HyperInverter (Dinh et al., 2022), FeatureStyleEncoder (FSE) (Yao et al., 2022) and StyleRes (Pehlivan et al., 2023), which benefit from multi-stage and multi-level information. Literately, we used the official pre-trained weights and configurations released by the authors to perform our evaluation experiments. For the Stanford Car dataset and LSUN Church dataset, several methods are omitted from comparisons when the models were not released. When evaluating the editing and manipulation, another important criterion for GAN inversion, we run extensive experiments leveraging InterfaceGAN (Shen et al., 2020a) for the human face images and GANSpace (Härkönen et al., 2020) for the car and church images to ensure diverse editing directions. More specifically, for the face images, we adopted the edit direction from the previous method (Yao et al., 2022), using the smiling, eyeglasses and heavy makeup boundaries trained by InterFaceGAN. For the car and church images, we computed PCA directions following the official GANSpace implementation (Härkönen et al., 2020).

**Implementation details** In our experiments, the pretrained StyleGAN generator was directly sourced from the StyleGAN2 repository (Karras et al., 2020). We then employed the same pretrained encoder as a fixed component for implementing the projector. We adopted the backbone design from FSE (Yao et al., 2022). Then, we followed the previous encoder-based methods (Richardson et al., 2021; Tov et al., 2021; Alaluf et al., 2021), with the Ranger optimizer, which combined the Lookahead (Zhang et al., 2019) and the Rectified Adam (Liu et al., 2019) optimizer for training. we set the learning rate and other parameters as $l_r = 0.0001, \beta_1 = 0.95, \beta_2 = 0.999$. To guarantee the image domain not varying a lot for some special point, we set the $\lambda_{o1} = 0.8$ and $\lambda_{o2} = 0.1$ for the image domain loss. All evaluation experiments were conducted using a single NVIDIA GeForce RTX 4090 GPU.

*Figure 4.* Results of manipulation inversion across multiple editing directions on human faces. The notations including ± indicates that the image is first edited by the manipulation in the latent generating space, followed by the inversion to restore the original input.

## 5.2. Comparisons on Manipulation Inversion

As mentioned in this paper, the inversion of manipulation can well reflect the realism of inversion. We first compared our method for the manipulation inversion and the results are reported in Table 1. As can be seen from this table, our method significantly outperforms other approaches in terms of manipulation inversion across both domains, demonstrating substantial improvements in all evaluation metrics. Additional comparisons on editing realism are provided in Appendix-A. We further show subjective results in Fig. 4, whereas our method exhibits superior performances during manipulation inversion and achieves the best manipulation inversion accuracy. Additional qualitative results can be found in Appendix-B.

## 5.3. Comparisons on Reconstruction Accuracy

As proved in the Sec. 4.1, the manipulation inversion optimises both the reconstruction and editing aspects. We thus systematically conduct a series of experiments to underscore the alignment between manipulation inversion and existing evaluation metrics of GAN inversion, substantiating their compatibility and effectiveness. More specifically, we conducted experiments on different challenging scenarios which are the *de facto* choice to evaluate the performances for GAN inversion tasks. Fig. 6 illustrates the samples of our reconstruction results and the comparison with existing baseline methods is provided in Table 2. Again, our method outperforms other encoder-based methods for reconstruction accuracy in all scenarios, exhibiting the superior perfor-

*Table 2.* Evaluations on the reconstruction accuracy. The best performance is highlighted in *red* and the second-best in *blue*.

| Method | MSE ↓ | LPIPS ↓ | SSIM ↑ | MS-SSIM ↑ |
|---|---|---|---|---|
| pSp (Richardson et al., 2021) | 0.0497 | 0.2790 | 0.6218 | 0.7208 |
| E4E (Tov et al., 2021) | 0.0663 | 0.3510 | 0.5985 | 0.6812 |
| ReStyle$_{pSp}$ (Alaluf et al., 2021) | 0.0401 | 0.2050 | 0.6394 | 0.7613 |
| ReStyle$_{E4E}$ (Alaluf et al., 2021) | 0.0600 | 0.3232 | 0.6015 | 0.7032 |
| HyperInverter (Dinh et al., 2022) | 0.0256 | 0.1481 | 0.6722 | 0.8105 |
| HFGI (Wang et al., 2022) | 0.0445 | 0.1803 | 0.6937 | 0.7495 |
| FSE (Yao et al., 2022) | *0.0215* | *0.0990* | *0.7550* | *0.8678* |
| E2Style (Wei et al., 2022) | 0.0481 | 0.3037 | 0.6595 | 0.7591 |
| **Ours** | *0.0133* | *0.0810* | *0.7670* | *0.8971* |

mances of our method that focuses on aligning distributions in the latent space to invert the arbitrary manipulation.

## 5.4. Ablation Study

To demonstrate the impact of each component in our method, we conducted step-by-step experiments to validate the effectiveness of the additional metrics and operations within the latent space. The MSE constraint on the latent space is referred to as the *latent restriction*. We denote $\boldsymbol{\Sigma} = \mathbf{J}^T\mathbf{J}$ as the *Jacobian component* and $\boldsymbol{\Sigma} = \mathbf{S}^T\mathbf{S}$ as the *semantic component*. The term *Adversarial learning* refers to the adversarial training proposed in Sec. 4.3. We report the results in Table 3. Notably, the latent restriction alone significantly improves the manipulation inversion results, as evidenced by the LPIPS metric and other metrics. Furthermore, the Jacobian component introduces additional constraints, effectively aligning transformations in both the image and latent domains. This alignment leads to substantial improvement in the metrics, indicating enhanced consistency and accu-

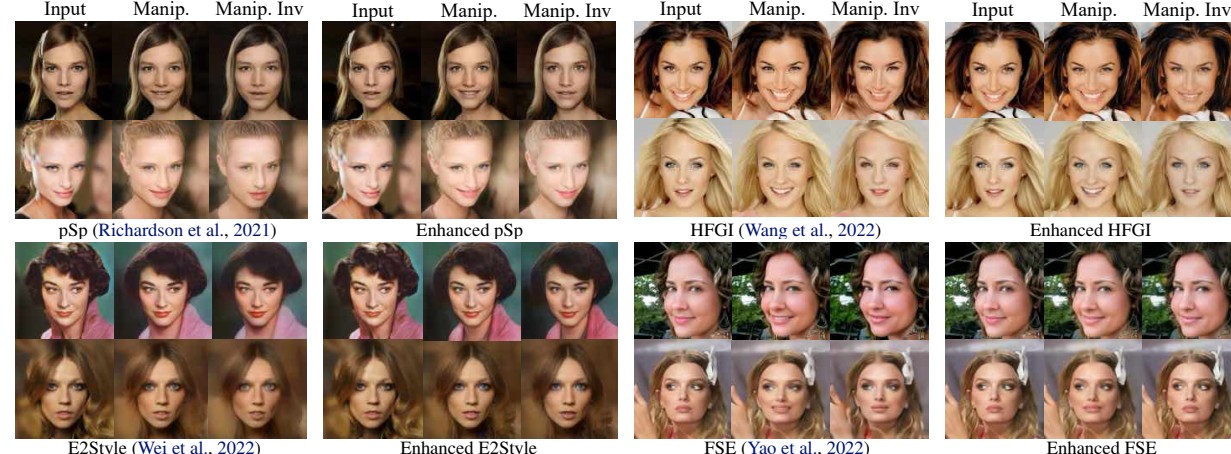

*Figure 5.* Comparisons between existing typical architectures and those enhanced by our method, including pSp (Richardson et al., 2021), HFGI (Wang et al., 2022), E2Style (Wei et al., 2022), and FSE (Yao et al., 2022). For each architecture, we show the input image, the manipulated image, and the inversion result. The enhanced versions demonstrate the improvements achieved by integrating our method.

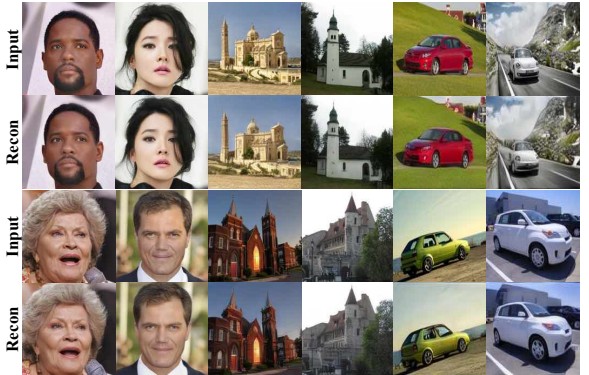

*Figure 6.* Illustration on the reconstruction accuracy of our method for GAN inversion.

racy in the inversion process. Our method, by sequentially incorporating all the components, consistently improves the inversion accuracy of manipulation, thus proving the effectiveness of each component.

*Table 3.* Ablation study evaluations on manipulation inversion against latent restriction, Jacobian component, semantic component and adversarial learning, which are 4 key components in the proposed method.

|  | MSE ↓ | LPIPS ↓ | SSIM ↑ | MS-SSIM ↑ |
|---|---|---|---|---|
| **Baseline** | 0.0223 | 0.1839 | 0.7115 | 0.8625 |
| + Latent restriction | 0.0212 | 0.1491 | 0.7217 | 0.8678 |
| + Jacobian component | 0.0154 | 0.1277 | 0.7354 | 0.8694 |
| + Semantic component | 0.0144 | 0.1265 | 0.7403 | 0.8926 |
| **+ Adversarial learning** | **0.0139** | **0.1263** | **0.7414** | **0.8931** |

### 5.5. Compatibility on Different Architectures

To demonstrate the universal property of our method, especially for the applicability across various encoder-based methods, we integrated different encoder types into our experiments, including a simple latent encoder (i.e., pSp), a two-phase encoder utilizing the shallow feature (i.e., HFGI), a multi-stage method incorporating the shallow feature (i.e., E2Style) and the state-of-the-art result (i.e., FSE). Our distri-

*Table 4.* The results of existing architectures enhanced by our method. The best results are highlighted in **Bold**.

|  | MSE ↓ | LPIPS ↓ | SSIM ↑ | MS-SSIM ↑ |
|---|---|---|---|---|
| pSp (Richardson et al., 2021) | 0.0500 | 0.3656 | 0.5875 | 0.7196 |
| **Enhanced pSp** | **0.0478** | **0.3473** | **0.5934** | **0.7290** |
| HFGI (Wang et al., 2022) | 0.0446 | 0.3198 | 0.5817 | 0.7481 |
| **Enhanced HFGI** | **0.0315** | **0.2618** | **0.6201** | **0.7905** |
| E2Style (Tov et al., 2021) | 0.0481 | 0.4148 | 0.6253 | 0.7590 |
| **Enhanced E2Style** | **0.0453** | **0.3497** | **0.6271** | **0.7650** |
| FSE (Yao et al., 2022) | 0.0223 | 0.1839 | 0.7115 | 0.8625 |
| **Enhanced FSE** | **0.0139** | **0.1263** | **0.7414** | **0.8931** |

bution estimation training was systematically applied to re-align the latent code within their respective latent spaces. All the training strategies were the same, with the exception of HFGI. Given the fact that HFGI refines images exclusively in the second stage, we adapt its methodology by initially training an E4E encoder, followed by training the second-stage consultation encoder using the default procedure of HFGI. The results are reported in Table 4, which exhibits the consistent improvements when using our method. Additionally, subjective results in Fig. 5 illustrate the effectiveness of integrating our method across different architectures.

## 6. Conclusion

In this paper, we have systematically analysed the latent generating space of generative adversarial network (GAN), by realising that the local curvature exists when inverting images. Motivated by this, we have proposed a new strategy, namely, inverting manipulations, instead of inverting images, by modelling the latent space as probabilistic models, and corespondingly establishing the statistical manifold. We then further proposed an adversarial training method to achieve efficient optimisation on calculating the manipulation inversion loss. The proposed method can flexibly act as plugin method to improve the inversion performances on different architectures. Experimental results have demonstrated superior performances on both reconstruction accuracy and editing realism.

## Broader Impact

The proposed method bridges GAN inversion and statistical manifold theory to unify reconstruction accuracy and editing realism, offering a novel perspective for high-fidelity image manipulation. By establishing a latent statistical manifold and adversarial optimization, our framework serves as a universal plugin for diverse GAN architectures, enabling seamless integration into applications such as medical imaging restoration, artistic content generation, and video compression. This universality reduces the need for architecture-specific adaptations, broadening its adoption in cross-domain tasks.

Furthermore, our adversarial strategy for minimizing manipulation inversion metrics introduces a computationally efficient paradigm for latent space optimization. This could inspire future research in unsupervised representation learning, particularly in scenarios requiring robustness to semantic perturbations, such as domain adaptation or anomaly detection.

However, enhanced editing realism may lower the barrier for generating deceptive content (e.g., deepfakes). To mitigate misuse risks, we advocate for ethical guidelines and detection frameworks to accompany such advancements. Future work should explore embedding traceability mechanisms within the latent space and fostering public awareness of synthetic media risks. By balancing innovation and responsibility, our method aims to advance generative technologies while safeguarding societal trust.

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

## A. Edit Realism Result

Our experimental evaluations were performed based on various scenarios. For the widely tested human face scenarios, we evaluated based on the CelebA-HQ (Karras et al., 2018) dataset for the inversion. We also evaluated our method for the car scenario, based on the $512 \times 512$ images within the Stanford Cars (Krause et al., 2013) to serve for training and evaluation with the official split.

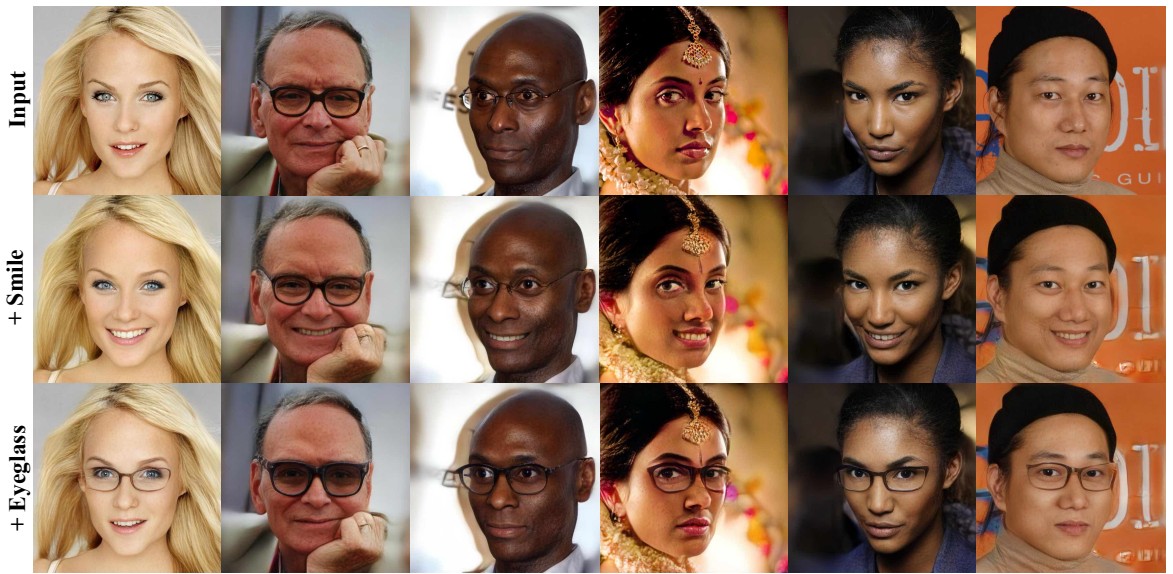

*Figure 7.* Samples of edit results in face domain on the CelebA-HQ (Karras et al., 2018) dataset.

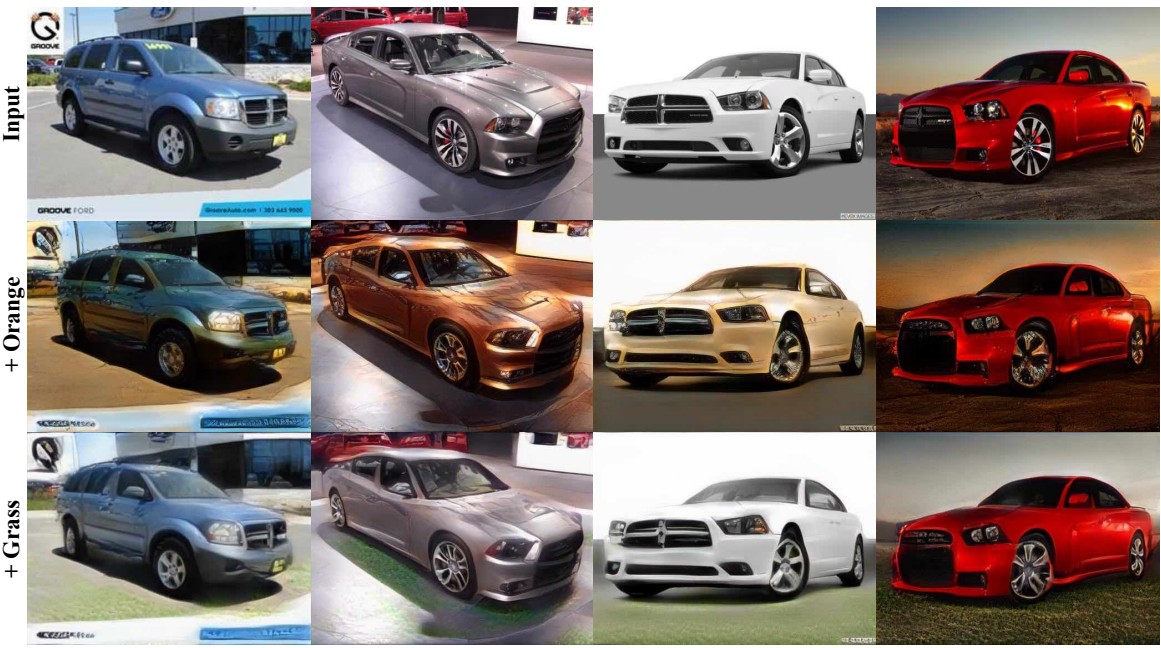

*Figure 8.* Samples of edit results in car domain on the Stanford Cars (Krause et al., 2013) dataset.

As proved in the paper, the manipulation inversion optimises both the reconstruction and editing aspects. The inversion of manipulation can well reflect the realism of inversion. We first exhibiting the comparisons on editing realism result. The editing realism result are shown in the Fig. 7 and 8. We show two edit directions of each image domain. For the face domain, we use the editing direction from InterfaceGAN (Shen et al., 2020b), for the car domain, we adapt the direction from GANSpace (Härkönen et al., 2020).

## B. Manipulation Inversion Result

We further show subjective results in Fig. 9, where our method exhibits superior stability during manipulation inversion and achieves the best reconstruction.

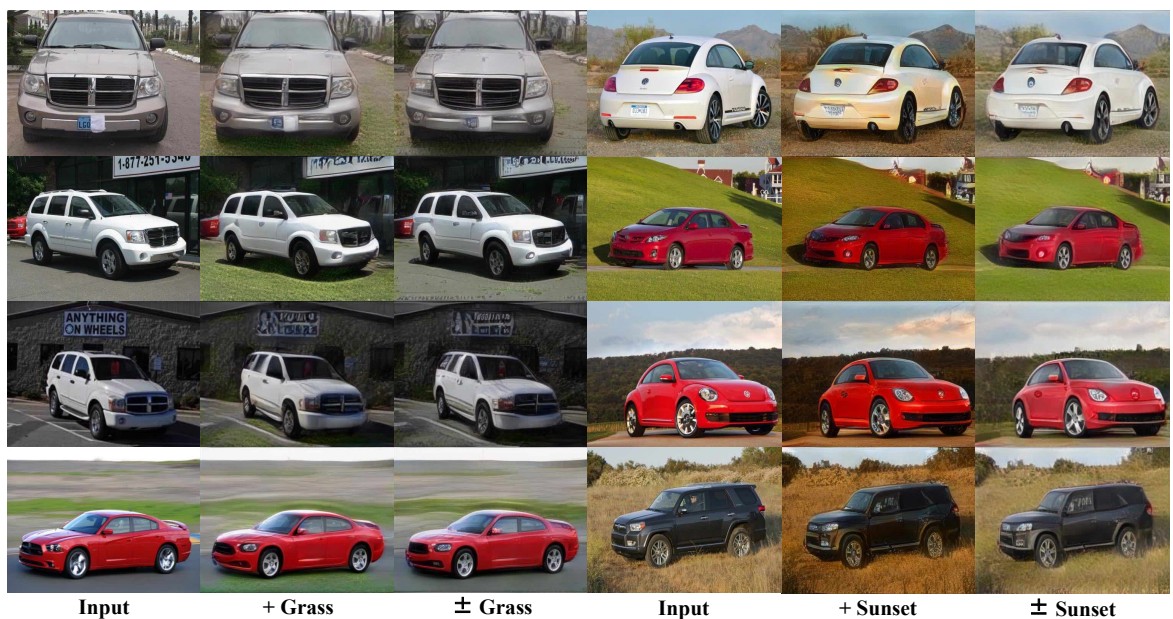

| Input | + Grass | ± Grass | Input | + Sunset | ± Sunset |

*Figure 9.* Manipulation inversion results in car domain on the Stanford Cars (Krause et al., 2013) dataset.

## C. Proof of Lemma 4.2

The proof of the Lemma 4.2 is listed below.

**Lemma C.1** (Lemma 4.2). *Let $g(\cdot)$ denote the pre-trained generator, and $f(\cdot)$ to represent the inversion encoder. Given an arbitrary latent feature $\mathbf{w}$ from an image $\mathbf{w} = f(\mathbf{x})$ and direction $\mathbf{v} \in \mathcal{B}_\epsilon(\mathbf{w})$, where $\mathcal{B}_\epsilon(\mathbf{w})$ represents an open ball of $\mathbf{w}$ with radius $\epsilon$, we represent the edited image by $\widetilde{\mathbf{x}} = g(\mathbf{w} + \mathbf{v})$. Then, the arbitrarily edited image $\widetilde{\mathbf{x}}$ can be precisely inverted, i.e., $\widetilde{\mathbf{x}} = g(f(\widetilde{\mathbf{x}}))$, if and only if we are able to invert the manipulation, i.e., $f(\widetilde{\mathbf{x}}) - \mathbf{v} = f(\mathbf{x})$.*

*Proof.* We can prove the equivalence between inverting arbitrarily edited images and inverting manipulation, through sufficiency and necessity.

- Sufficiency: If any edited images can be inverted, we have $\widetilde{\mathbf{x}} = g(f(\widetilde{\mathbf{x}}))$ for any $\mathbf{v} \in \mathcal{B}_\epsilon(\mathbf{w})$. On the other hand, $\widetilde{\mathbf{x}}$ is generated by $\widetilde{\mathbf{x}} = g(\mathbf{w} + \mathbf{v})$. Then, we arrive at

$$g(f(\widetilde{\mathbf{x}})) = \widetilde{\mathbf{x}} = g(\mathbf{w} + \mathbf{v}).$$

  As the generator $g$ is continuous and acts as injection mapping, we can safely remove $g(\cdot)$ and thus have

$$f(\widetilde{\mathbf{x}}) = \mathbf{w} + \mathbf{v}$$

  Therefore, we prove the sufficiency $f(\mathbf{x}) = \mathbf{w} = f(\widetilde{\mathbf{x}}) - \mathbf{v}$ for any $\mathbf{v} \in \mathcal{B}_\epsilon(\mathbf{w})$.

- Necessity: Given $f(\widetilde{\mathbf{x}}) - \mathbf{v} = f(\mathbf{x})$ for any $\mathbf{v} \in \mathcal{B}_\epsilon(\mathbf{w})$, we thus have

$$\widetilde{\mathbf{x}} = g(f(\widetilde{\mathbf{x}})) = g(f(\mathbf{x}) + \mathbf{v})$$

  which obtains $\widetilde{\mathbf{x}} = g(\mathbf{w} + \mathbf{v})$. This proves that any edited image can be precisely inverted.

This completes the proof. □