# OpenReview forum: "Manipulation Inversion by Adversarial Learning on Latent Statistical Manifold"
_ICML.cc/2025/Conference — Submitted to ICML 2025_

### Official Review · Reviewer_jjf2 · 2025-03-14

**Overall Recommendation:** 4

**Summary:**

This paper aims at improving the gan inversion method to achieve both good reconstruction and realism of editing. Several findings from the paper indicates that to really invert an image back to the latents space, it better to prevent from finding only the local minimum (which harms the realism of editing), but to also preserve the manifold of the the latent space.

**Claims And Evidence:**

I find the several findings from section 3 of the paper quite persuasive and interesting.

However, I am not fully convinced by the performance of this paper. Although claimed by the authors, better manipulation inversion is equal to better inversion, it is how well the editing after inversion that matters. There are enough evidence that the method performs better than others in manipulation inversion, yet I can not find clear evidence that this paper also perform better in editing realism.

**Essential References Not Discussed:**

Not to my knowledge.

**Experimental Designs Or Analyses:**

The given experiments looks good.

**Methods And Evaluation Criteria:**

The proposed method make sense, as the previous point-based inversion methods does not guarantee a good editing after inversion.
The metrics for inversion are reasonable. But as I mentioned above, not enough metrics about editing are presented.

**Other Comments Or Suggestions:**

No.

**Other Strengths And Weaknesses:**

1. Strengths - This paper attempts to solve the general trade-off between inversion accuracy and editing performance in GAN. The findings in the paper and the proposed method is quite interesting and make sense to me.
2. Strengths - This paper shows better performance in manipulation editing.

1.  Weaknesses- As I mentioned in "Claims And Evidence", more comparisons are needed to show the improvement in this paper in editing.
2.  Weaknesses- I think there is an important experiment for the authors to complete: given a ground truth latent code and the generated image, when inverted and get a inverted latent, how similar are the gt latent and inverted latent? The similarity can be measured in both distance, and how they would response given the same editing (this should be what this paper perform better than the baselines.)
3. Weaknesses- The writing in section 4 is hard to follow. I understand the objective of  separate equations, yet the overall pipeline is still unclear to me. For example, the L_{j} and L_{s} from eqn (9) are not introduced in the paper. Also, the computation of S and J in eqn (5) need further discussion other than referring to other paper.

**Questions For Authors:**

I will increase my score if the following questions can be answered.
1. Better writing for the method part. What's the entire pipeline? What's the meaning of the missing terms? See the "Claims And Evidence" for more detail.
2. More evaluation on inversion accuracy and editing performance. Please see the  "Other Strengths And Weaknesses" and "Claims And Evidence" for more detail.

**Relation To Broader Scientific Literature:**

Not to my knowledge.

**Theoretical Claims:**

The main assumption seems to be lemma 4.2. I think it I valid given the assumption holds.

---

> ### Author Rebuttal · Authors · 2025-04-01
>
> We wish to sincerely thank the reviewer for the valuable comments and insightful suggestions.
>
> **[Q1 Weakness 1: Editing Performance]**
>
> Indeed, as pointed out by Lemma 4.2 of our manuscript, the performance of editing is in accordance with the accuracy of manipulation inversion. Our best performances on manipulation inversion thus verify the superiority for image editing of our method. We also newly conduct extensive evaluations on image editing, in which our method consistently achieves the best editing performances. We refer to our response to Reviewer#1qCg [Q3] for more details.
>
> **[Q2 Weakness 2: Experiment of Inverting $w$]**
>
> Yes. We agree with the reviewer that the distance between the ground-truth and inverted latent codes is also crucial to evaluate the effectiveness of our method. As illustrated in [Figure 2](https://anonymous.4open.science/r/icml2025_14926/fig_w.pdf), we thus conducted new evaluations regarding the latent space MSE, and report the results in [Table 4](https://anonymous.4open.science/r/icml2025_14926/table_w.pdf). As can be seen from this table, our method, by effectively optimizing the manipulation inversion with our VAT strategy, consistently achieves the best accuracy. We further evaluate the MSE between editing images, by choosing the smile direction, and also report the results in [Table 4](https://anonymous.4open.science/r/icml2025_14926/table_w.pdf), which again achieves the superior performances.
>
> **[Q3 Weakness 3: Clarifying Section 4]**
>
> Indeed, the random manipulation is sampled by our VAT strategy proposed in Section 4, in which one manipulation direction is optimized at each iteration. More specifically, for each latent code $\mathbf{w}$, we compute the perturbation loss $\psi(\mathbf{w}, \mathbf{v})$, and iteratively solve for the worst-case direction $\mathbf{v}^*$ via power iteration on the Hessian of $\psi$, i.e., (Golub \& der Vorst, 2000) of our manuscript, ensuring maximal disruption to the inversion consistency. The worst-case direction $\mathbf{v}^*$ is then used to optimize our encoder to achieve the manipulation inversion. We further provide [Algorithm 1](https://anonymous.4open.science/r/icml2025_14926/alg.pdf) to depict the overall pipeline of our method.
>
> Moreover, $\mathcal{L}\_{j}$ and $\mathcal{L}\_{s}$ are the losses that correspond to the semantic and Jacobian terms. More specifically, by combining Equation (6) and (8) in our manuscript, we are able to calculate the Riemannian gradient given the target loss $\psi(\mathbf{w},\mathbf{v})=\mathbf{d}^T\mathbf{d}$, where $\mathbf{d}=f(g(\mathbf{w}+\beta\frac{\mathbf{v}}{||\mathbf{v}||_2}))-\beta\frac{\mathbf{v}}{||\mathbf{v}||_2}-\mathbf{w}$. In this way, we are able to move a step further based on Equation (6) of our manuscript, as follows
>
> $$
> \nabla_r\phi(\mathbf{w})=\nabla_e\phi(\mathbf{w})^T(\mathbf{S}^T\mathbf{S}+\mathbf{J}^T\mathbf{J})=2\mathbf{d}^T(\mathbf{S}^T\mathbf{S}+\mathbf{J}^T\mathbf{J}),
> $$
>
> where $\phi(\mathbf{w})=\psi(\mathbf{w},\mathbf{v})$. Therefore, the achieved Riemannian gradient via our established manifold is equivariant to calculate the Euclidean gradient by the modified target loss $\psi(\mathbf{w},\mathbf{v})=\mathbf{d}^T(\mathbf{S}^T\mathbf{S})\mathbf{d}+\mathbf{d}^T(\mathbf{J}^T\mathbf{J})\mathbf{d}$, and we thus denote $\mathcal{L}\_{j}=\mathbf{d}^T(\mathbf{J}^T\mathbf{J})\mathbf{d}$ and $\mathcal{L}\_{s}=\mathbf{d}^T(\mathbf{S}^T\mathbf{S})\mathbf{d}$, such that the manipulation inversion loss can be effectively optimized via our final loss given by Equation (9) of our manuscript.
>
> Furthermore, $\mathbf{J}$ is essentially the Jacobian of the pre-trained generator, which can be computed efficiently by the gradients during each iteration, namely, via backpropagation with respect to pixel outputs (following Ramesh et al., 2018). On the other hand, $\mathbf{S}$ consists of semantic directions, obtained by either supervised or unsupervised manners, which formulates the semantic linear space in the latent space; this is obtained by Shen et al., 2020. We wish to thank the reviewer for pointing out the ambiguity of our Section 4, and we will elaborate more on this, including the calculation on $\mathbf{S}$ and $\mathbf{J}$, in the revision.

---

> > ### Comment · Reviewer_jjf2 · 2025-04-06
> >
> > I appreciate the authors for the rebuttal.
> > I will increase my rating, with the belief that the author will improve their writing, adding the more qualitative and quantitative evaluations in the final version.

---

> > > ### Author Response · Authors · 2025-04-07
> > >
> > > Thank you very much for your positive opinion and valuable comments! In the revised version, we will further improve our writing, together with comprehensively including more qualitative and quantitative evaluations to solid evaluations of our advantages.

---

### Official Review · Reviewer_3REN · 2025-03-14

**Overall Recommendation:** 2

**Summary:**

This paper aims to enhance the editing ability of current GAN inversion methods. First, this paper investigates the properties of the latent space of StyleGAN and obtains three interesting findings (Sec. 3). Based on these findings, this paper proposes adversarial learning for latent manipulation inversion and anisotropic Gaussian distribution for latent features. Experimental results show improved reconstruction quality in latent manipulation inversion.

**Claims And Evidence:**

The quality of editing pertains not only to the preservation of image details that are not intended for modification but also to the accuracy/quality of the desired attribute (direction). I believe that a more effective manipulation inversion can enhance detail preservation. However, I do not think it necessarily improves the accuracy/quality of the target attributes.

**Essential References Not Discussed:**

The essential related works are well-discussed and cited.

**Experimental Designs Or Analyses:**

This paper does not evaluate how the proposed method affects the quality and accuracy of the editing results.

**Methods And Evaluation Criteria:**

No, this paper does not evaluate how the proposed method affects the quality and accuracy of the editing results. For example, does the proposed method reduce the accuracy of adding or removing glasses?

**Other Comments Or Suggestions:**

N/A

**Other Strengths And Weaknesses:**

### Strengths
+ A thorough analysis of the latent space in StyleGAN, with potential generalization to broader latent spaces, such as the h-space in Diffusion Models.

### Weaknesses
- There is a lack of a comprehensive study on how the proposed method affects editing accuracy and quality. For example, the Fréchet Inception Distance (FID) score of the edited results and the accuracy of adding or removing an attribute. These metrics are commonly used in other studies on image editing.
- I do not see the definitions of $L_j$ and $L_s$ in (9).

**Questions For Authors:**

1. As mentioned above, please discuss or evaluate how the proposed method affects the editing quality/accuracy.
2. Please provide the definitions of $L_j$ and $L_s$.

**Relation To Broader Scientific Literature:**

The findings of StyleGAN's latent space are interesting; they provide a deeper understanding of StyleGAN and GAN inversion. The analytical method for these findings can be generalized to more common latent spaces.

**Theoretical Claims:**

Yes.

---

> ### Author Rebuttal · Authors · 2025-04-01
>
> Many thanks for the insightful suggestions.
>
> **[Q1 Weakness 1: Quality and Accuracy of the Editing Results]**
>
> Indeed, the realism and accuracy of image editing are tightly related to the accuracy of manipulation inversion, which has been proved in our manuscript. We further conducted comprehensive evaluations on the editing performances. We may refer to our response to Reviewer#1qCg [Q3] for more details. Notably, for several target attributes with distinct semantics, we calculate the Clip scores to assess the semantic accuracy of edited images, in which our method achieves superior performances. Our method also obtains the lowest FIDs and highest ClipIQA scores, verifying the superior quality of edited images. This is due to the manipulation inversion oriented optimization, in which advanced domains are searched and optimized in the latent space of StyleGAN.
>
> **[Q2 Weakness 2: Clarifying $\mathcal{L}_j$ and $\mathcal{L}_s$]**
>
> Many thanks for pointing out this. Indeed, by combining Equation (6) and (8) in our manuscript, we are able to calculate the Riemannian gradient given the target loss $\psi(\mathbf{w},\mathbf{v})=\mathbf{d}^T\mathbf{d}$, where $\mathbf{d}=f(g(\mathbf{w}+\beta\frac{\mathbf{v}}{||\mathbf{v}||_2}))-\beta\frac{\mathbf{v}}{||\mathbf{v}||_2}-\mathbf{w}$. In this way, we are able to move a step further based on Equation (6) of our manuscript, as follows
>
> $$
> \nabla_r\phi(\mathbf{w})=\nabla_e\phi(\mathbf{w})^T(\mathbf{S}^T\mathbf{S}+\mathbf{J}^T\mathbf{J})=2\mathbf{d}^T(\mathbf{S}^T\mathbf{S}+\mathbf{J}^T\mathbf{J}),
> $$
>
> where $\phi(\mathbf{w})=\psi(\mathbf{w},\mathbf{v})$. Therefore, the achieved Riemannian gradient via our established manifold is equivariant to calculate the Euclidean gradient by the modified target loss $\psi(\mathbf{w},\mathbf{v})=\mathbf{d}^T(\mathbf{S}^T\mathbf{S})\mathbf{d}+\mathbf{d}^T(\mathbf{J}^T\mathbf{J})\mathbf{d}$, and we thus denote $\mathcal{L}\_{j}=\mathbf{d}^T(\mathbf{J}^T\mathbf{J})\mathbf{d}$ and $\mathcal{L}\_{s}=\mathbf{d}^T(\mathbf{S}^T\mathbf{S})\mathbf{d}$, such that the manipulation inversion loss can be effectively optimized via our final loss given by Equation (9) of our manuscript.

---

### Official Review · Reviewer_oWVi · 2025-03-15

**Overall Recommendation:** 4

**Summary:**

This article introduces a manipulation inversion method for GAN models. It constructs the generative manifold using different editing vectors to create a more stable and reliable inversion space.

**Claims And Evidence:**

This article conducts extensive experiments to demonstrate that their method achieves state-of-the-art (SOTA) performance. However, based on their theory, the approach could potentially work with multiple directions, though this is not explicitly stated.

**Essential References Not Discussed:**

[1] Bhattad, Anand, et al. "Make it so: Steering stylegan for any image inversion and editing." arXiv preprint arXiv:2304.14403 (2023).

[2] Wang, Tengfei, et al. "High-fidelity gan inversion for image attribute editing." Proceedings of the IEEE/CVF conference on computer vision and pattern recognition. 2022.

[3] Yao, Xu, et al. "Feature-style encoder for style-based gan inversion." arXiv preprint arXiv:2202.02183 (2022).

[4]Roich, Daniel, et al. "Pivotal tuning for latent-based editing of real images." ACM Transactions on graphics (TOG) 42.1 (2022): 1-13.

**Experimental Designs Or Analyses:**

In inversion research, the primary focus should be on the editing of inverted images. However, there is limited experimental work addressing this aspect. For face images, you can compute the ID metric [1] to demonstrate that your method preserves identity consistency between the original, inverted, and edited images.
Also, you should show the time and space using in your method

[1]Jiankang Deng, Jia Guo, Niannan Xue, and Stefanos Zafeiriou. ArcFace: Additive angular margin loss for deep face recognition. In Proceedings of the IEEE/CVF conference on computer vision and pattern recognition, pages 4690–4699, 2019.

**Methods And Evaluation Criteria:**

This research is helpful to improve the quality of the image inversion.

**Other Comments Or Suggestions:**

no

**Other Strengths And Weaknesses:**

Strengths:
Introduce a new method to inverse images and the results are more stable and reliable.
This article is good writing.

Weakness:
please see the questions.

**Questions For Authors:**

Initially, the goal is to use different directions to construct the estimated manifold. However, due to the high dimensionality of the latent space, it is challenging to sample all possible directions or optimize the space manifold effectively.

Secondly, in equation (7) of section 4.3, the objective is to maximize 𝑣 , but since 𝑣 is a vector, its meaning in this context is unclear. If the intention is to maximize ∣∣𝑣∣∣, there is a discrepancy because the equation normalizes 𝑣 to 𝑣/∣∣𝑣∣∣, making it a unit vector that does not influence the maximization directly.

Thirdly, there are some other method you should compare with:

[1] Bhattad, Anand, et al. "Make it so: Steering stylegan for any image inversion and editing." arXiv preprint arXiv:2304.14403 (2023).

[2] Wang, Tengfei, et al. "High-fidelity gan inversion for image attribute editing." Proceedings of the IEEE/CVF conference on computer vision and pattern recognition. 2022.

[3] Yao, Xu, et al. "Feature-style encoder for style-based gan inversion." arXiv preprint arXiv:2202.02183 (2022).

[4]Roich, Daniel, et al. "Pivotal tuning for latent-based editing of real images." ACM Transactions on graphics (TOG) 42.1 (2022): 1-13.

Forthly, when people inite the point on the high density part, like hubness latents [5], it will fail to inverse the images. Would your method have the similar problem with it?

[5] Liang, Yuanbang, et al. "Exploring and exploiting hubness priors for high-quality GAN latent sampling." International Conference on Machine Learning. PMLR, 2022.

Finally, while your claims suggest that using multiple directions could be effective, this approach is not reflected in your experimental results.

**Relation To Broader Scientific Literature:**

in the previous method, it is one-way computing. This method is to multi-inverse optimizing the encoder manifold.

**Theoretical Claims:**

This article is to claim they apply the w'=w+v to re-encode and mimax the similarity of the results and the v the manifold space. But, after reading, there are some issues should be disscussed:
1. the goal is to use different directions to construct the estimated manifold. However, due to the high dimensionality of the latent space, it is challenging to sample all possible directions or optimize the space manifold effectively.

2. in equation (7) of section 4.3, the objective is to maximize 𝑣 , but since 𝑣 is a vector, its meaning in this context is unclear. If the intention is to maximize ∣∣𝑣∣∣, there is a discrepancy because the equation normalizes 𝑣 to 𝑣/∣∣𝑣∣∣, making it a unit vector that does not influence the maximization directly.

---

> ### Author Rebuttal · Authors · 2025-04-01
>
> Many thanks for the insightful comments.
>
> **[Q1 Theoretical Claims: Multiple Directions]**
>
> **Questions for Authors 1: Handling Multiple Directions in Manifold Construction:** Indeed, our manifold is established based on the semantic direction $\mathbf{S}$ and Jacobian matrix $\mathbf{J}$. The Jacobian matrix $\mathbf{J}$ is defined by excessive sampling in the latent space. However, we effectively employed the gradients of the generator to obtain $\mathbf{J}$, following (Ramesh et al., 2018) of our manuscript; this avoids the excessive sampling procedure. Our manipulation inversion requires sampling $\mathbf{v}$ when calculating the loss of Equation (1) in our manuscript, which is intractable. We thus propose the VAT strategy, by choosing the worst-case $\mathbf{v}^*$. The worst-case $\mathbf{v}^*$ can also be effectively solved by power iteration on the Hessian of $\psi$, i.e., (Golub & der Vorst, 2000) of our manuscript. We also provide our detailed pipeline in the newly added [Algorithm 1](https://anonymous.4open.science/r/icml2025_14926/alg.pdf).
>
> **Questions for Authors 2: Clarification on Equation (7) and Maximization of $\mathbf{v}$:** Eq. (7) aims to maximize the perturbation loss $\psi(\mathbf{w},\mathbf{v})$, by choosing the worst-case $\mathbf{v}^*$. By constraining $\mathbf{v}$ to a unit vector ($\frac{\mathbf{v}}{||\mathbf{v}||_2}$), we enforce the maximization by focusing on directions, instead of infinitely increasing the scales in the latent space. The scalar $\beta$ governs the perturbation strength instead. This way, the resulting $\mathbf{v}^*$ identifies the direction that most disrupts inversion consistency. This way, our VAT-related loss in Equation (7) of our manuscript essentially formulates an upper bound of the primary manipulation inversion loss given by Equation (1) of our manuscript. Based on this, minimizing  $\psi(\mathbf{w},\mathbf{v}^*)$ thus ensures the robust and effective convergence of our encoder $f$ to arbitrary perturbations.
>
> **[Q2 Experimental Designs Or Analyses: Editing Performance and Method Complexity]**
>
> Indeed, as pointed out by Lemma 4.2 of our manuscript, the performance of editing is in accordance with the accuracy of manipulation inversion. Our best performances on manipulation inversion thus verify the superiority for image editing of our method. We also newly conduct extensive evaluations on image editing, in which our method consistently achieves the best editing performances. We refer to our response to Reviewer#1qCg [Q3] for more details.
>
> Regarding the complexity, instead of excessively sampling multiple directions, the proposed VAT strategy effectively reduces the computational complexity of our method. However, our method needs to calculate the Jacobian of the generator, thus requiring to forward twice during training. Consequently, our method was trained on a single NVIDIA GeForce RTX 4090 GPU, with a total training time of about $80$ hours. Regarding the FSE baseline, it consumed 50 hours as comparison. Both methods consumed comparable memory requirements. Inference with our method does not increase computational complexity or memory cost, compared to existing state-of-the-art GAN inversion methods.
>
> **[Q3 Questions for Authors 3: Comparing Methods]**
>
> Ref. [1] operates in Z space for better inversion accuracy and editing. However, we did not find the publicly available codes within the tight rebuttal period. Ref. [2] (HFGI) and Ref. [3] (FSE) are two state-of-the-art comparing methods and have been already compared in our manuscript. For Ref. [4], the proposed PTI is an optimization-based method, which fine-tunes the generator to allow for improved accuracy for each image, instead of the encoder-based methods for all images including Refs. [2,3] and our method. This requires excessive computation to search for the best per image during the inference. In the rebuttal, we also report the new comparisons of PTI in [Table 1](https://anonymous.4open.science/r/icml2025_14926/table_edit.pdf), [Table 2](https://anonymous.4open.science/r/icml2025_14926/table_id.pdf), and [Table 3](https://anonymous.4open.science/r/icml2025_14926/tab_reconstruction_quatitative_results.pdf), in which our method also achieves the best performances for both reconstruction and editing.
>
> **[Q4 Questions for Authors 4: Hubness Problem]**
>
> Indeed, we did not observe the hubness problem empirically. This may due to two possible aspects: (a) **Pretrained Encoder Initialization:** The fixed encoder maps inputs to semantically stable regions in latent space, avoiding hubness-prone initialization that commonly appears in random sampling. (b) **Multi-Constraint Optimization:** Jacobian regularization stabilizes gradients, while semantic constraints anchor optimization to plausible latent regions, jointly preventing convergence to hubness-dominated local optima. We believe further analysis on hubness problem could improve inversion, which is left for future work.

---

> > ### Comment · Reviewer_oWVi · 2025-04-03
> >
> > Thank you for your response. I appreciate the clarifications, but I still have a few questions:
> >
> > 1. In the updated Algorithm 1, there are some square symbols—could you clarify what they represent?
> >
> > 2. In #1qCg [Q3], it is mentioned that the ClipDiff score is defined. However, the results are not presented as they are in Algorithm 1 and Tables 1, 2, and 3. Could you provide more details or clarify this?
> >
> > 3. Would it be possible to include an example of hubness initialization in the final version? It sounds like a significant improvement, and I would appreciate a concrete demonstration.
> >
> > But I'm happy to improve the score.

---

> > > ### Author Response · Authors · 2025-04-07
> > >
> > > We thank the reviewer very much for the valuable comments and insightful suggestions. We are glad that we've addressed your questions!
> > >
> > > **[Q1 Clarifying Algorithm 1]**
> > > The square symbols in [Algorithm 1](https://anonymous.4open.science/r/icml2025_14926/alg.pdf) indicate squared $l_2$-norms. More specifically, the perturbation loss $\psi(\mathbf{w}, \mathbf{v}) = ||f(g(\mathbf{w} + \beta \frac{\mathbf{v}}{||\mathbf{v}||_2})) - \beta \frac{\mathbf{v}}{||\mathbf{v}||_2} - \mathbf{w}||_2^2$ measures the squared $l_2$ distance between the perturbed reconstruction $f(g(\mathbf{w} + \beta \frac{\mathbf{v}}{||\mathbf{v}||_2})) - \beta \frac{\mathbf{v}}{||\mathbf{v}||_2}$ and the original latent code $\mathbf{w}$, where the subscript $||\cdot||_2$ denotes the $l_2$ norm and the superscript $||\cdot||^2$ is the squaring operation. Similarly, the manipulation inversion loss $\psi(\mathbf{w}, \mathbf{v^*})$ also employs the squared $l_2$-norm to quantify reconstruction error from the encoder output.
> > >
> > > **[Q2 Clarifying ClipDiff Score]**
> > > Yes, we introduce ClipDiff to evaluate editing performance on the Church dataset, where ground-truth attribute directions are not available. We report the ClipDiff in [Table 1](https://anonymous.4open.science/r/icml2025_14926/table_edit.pdf), under the “Church Editing” section. This is due to unlike human faces with predefined semantic attributes (e.g., eyeglasses), Church dataset lacks ground-truth annotated attributes, in which the edit directions are obtained via GANSpace in an unsupervised style. Therefore, we cannot rely on traditional metrics such as ID and CLIP scores in [Table 2](https://anonymous.4open.science/r/icml2025_14926/table_id.pdf), in which ground-truth labels are required. On the other hand, [Table 3](https://anonymous.4open.science/r/icml2025_14926/tab_reconstruction_quatitative_results.pdf) reports the reconstruction accuracy, in which the ClipDiff score may not be suitable.
> > >
> > > To address this, we develop ClipDiff to evaluate editing effectiveness, which is defined as the cosine distance between CLIP image embeddings from the input and edited images. A larger ClipDiff score indicates a distinct semantic shift, capturing editing content. We also use ClipIQA as complementary to assess the perceptual quality of the edited images. This ensures that the editing performance is evaluated from being both semantically meaningful and visually coherent.
> > >
> > > We therefore report both ClipDiff and ClipIQA scores for the Church dataset in [Table 1](https://anonymous.4open.science/r/icml2025_14926/table_edit.pdf), under the “Church Editing” section. From this table, our method achieves the highest ClipDiff ($\uparrow0.4273$) and ClipIQA ($\uparrow0.5104$), indicating that our edits are both semantically distinct and perceptually superior quality than all the baselines. We shall further clarify our comparisons based on ClipDiff score in our revised version, together with comprehensive evaluations for editing and comparing methods.
> > >
> > > **[Q3 A Concrete Demonstration of Hubness Initialization]**
> > > We appreciate the insightful feedback! Following the suggestion, we conducted additional experiments on hubness latent features. Indeed, the input of our method is the real-world images, and we added new experiments on our inverted latent features from real-world images, acting as the initialization for the StyleGAN generator.
> > >
> > > We then calculated the portion of falling into the high-density regions, i.e., belonging to the hubness latent feature that deteriorates the inversion. More specifically, since the threshold $t$ determines the minimum number of $k$-nearest points for the current latent feature to be regarded as the hubness, we inverted the latent features from $10k$ test images and evaluated the numbers of hubness latent features under varying $t$  thresholds. We report the results in [Figure 3](https://anonymous.4open.science/r/icml2025_14926/fig_hubness.pdf) and [Table 5](https://anonymous.4open.science/r/icml2025_14926/table_hubness.pdf), in which the default setting of $t$ is 50 in the suggested Ref. [5]. As can be seen from this figure and table, our approach consistently results in the smallest numbers of hubness latent features across $10k$ samples and different $t$ thresholds, whereas baseline methods such as FSE, E2Style and pSp still exhibit considerable concentration in these problematic areas. For the default setting of $t=50$, our method achieves non-hubness latent features for all $10k$ samples. This statistically demonstrates that our encoder-driven mapping effectively avoids high-density regions—commonly referred to as hubness—in the $W$-space of StyleGAN, which negatively impact inversion quality as pointed out by the reviewer.
> > >
> > > In our revised version, we will further clarify this in our final version, emphasizing that the observed advantage is an natural property of our pretrained encoder initialization and manifold-assisted optimization on the latent space of StyleGAN.

---

### Official Review · Reviewer_1qCg · 2025-03-17

**Overall Recommendation:** 3

**Summary:**

The paper proposes Manipulation Inversion, a novel GAN inversion method that addresses the inherent trade-off between accurate image reconstruction and realistic editing by focusing on manipulating latent spaces rather than direct image reconstruction. Motivated by a systematic analysis of the latent space in StyleGAN, the authors uncover critical issues: multiple latent representations corresponding to similar identities, anisotropic semantic variations, and inconsistent image variations from latent edits. To tackle these, the authors establish a latent statistical manifold using a factorized probabilistic model, capturing local curvature and semantic directions in the latent space. An adversarial learning strategy is introduced to efficiently optimize inversion performance without excessive sampling. Experimental results demonstrate superior manipulation inversion and reconstruction accuracy, validating the method's compatibility with diverse GAN architectures.


## update after rebuttal
Thank the authors for more information provided in rebuttal. I think many of my concerns have been addressed, including the implementation details and quantitative editing performance. But the additional visual examples still don't show any advanced editing case except the current glasses, smile and makeup. So I would just maintain my current ratings of weak leaning toward acceptance.

**Claims And Evidence:**

The observations and analyses of existing issues of current algorithms are accurate and reasonable to me. I would suggest to present more visual examples of the current limitations and failure cases etc. Fig. 2 shows a framework but lacks details.

**Essential References Not Discussed:**

I think this paper has cited sufficient related references.

**Experimental Designs Or Analyses:**

I think the major issue of the experiments and results is that the editing performance is less evaluated compared to the reconstruction part. The paper claims the editing quality as the key issue to address but lacks related evaluations.
Only Fig. 4 in the main paper displays some editing results, while the effects of "smiling" or "makeup" are natually trivial. The second row of "eyeglasses" doesn't show clearly change compared to other methods.
Figs. 7 and 8 in the appendix show more results, while the total examples are still limited, and the effects are not significant to apply, and no comparisons are made.
No table is provided to measure the editing performance. Although it's not as easy as for reconstructions to measure since there is no ground truth, there are still some indirect metrics such as CLIP score between the edited images and the editing prompt.

The datasets used in the experiments are also limited on human faces and cars. StyleGANs are also originally trained on the bedroom, church etc. datasets. There are also many other third-party pre-trained StyleGANs to leverage, so that readers can understand visually better how the editing performance varies across scenarios and attributes.

**Methods And Evaluation Criteria:**

The idea of introducing a unit random manipulation to train the inverter is novel and effective. I'm wondering how this random manipulation is sampled. The paper mainly illustrate it mathematically but lacks implementation details.

The evaluation for reconstruction is well conducted, while that for editing is not good enough. Please see below "Experimental Designs Or Analyses" section.

**Other Comments Or Suggestions:**

I have no other comments.

**Other Strengths And Weaknesses:**

Please see above.

**Questions For Authors:**

I have no other questions.

**Relation To Broader Scientific Literature:**

In my understanding the idea of introducing a random unit manipulation to train an editing-plausible inverter might be able to be extended to using human feedback or reinforcement learning, compared to the discriminator used in this work.

**Theoretical Claims:**

I checked the proofs and formulas in the paper and didn't find issues. I think they're clearly listed and deduced and easy to follow.

---

> ### Author Rebuttal · Authors · 2025-04-01
>
> Many thanks for the valuable and insightful comments!
>
> **[Q1 Claims and Evidence: More Visual Examples on Analysis]**
>
> Yes. We have conducted further analysis on the generating latent space of StyleGAN, revealing the possible limitations for GAN inversion. More specifically, Figure 2-(a) of our manuscript essentially depicts our Finding 1, and we further add new visual examples in [Figure 1](https://anonymous.4open.science/r/icml2025_14926/fig_findings.pdf)-(a), which further highlights the multi-domain characteristics.
>
> Regarding our Finding 2, Figure 2-(b) of our manuscript emphasizes the anisotropy property within the latent space. We also provide more visual examples in [Figure 1](https://anonymous.4open.science/r/icml2025_14926/fig_findings.pdf)-(b), by additionally calculating the identity (ID) values. From this figure, we can conclude that for the same scale, different manipulation directions exhibit distinct anisotropy across IDs and semantics.
>
> Moreover, our Finding 3, together with Figure 3-(c) of our manuscript, reveals the inconsistent variation. We also clarify the details and provide more visual illustrations in [Figure 1](https://anonymous.4open.science/r/icml2025_14926/fig_findings.pdf)-(c), which demonstrate the non-monotonic trends of MSE in the image space, when consistently increasing the editing scale. Therefore, the above analysis reveals the characteristics of the latent space, in which existing GAN inversion methods may be limited during the optimization. For example, the majority of existing GAN inversion methods optimize the image MSE, which may not ensure the semantic consistency and accuracy within the latent space as pointed out by our Finding 3.
>
> **[Q2 Methods And Evaluation Criteria: Lack Implementation Details]**
>
> Indeed, the random manipulation is sampled by our VAT strategy, in which one manipulation direction is optimized at each iteration. More specifically, for each latent code $\mathbf{w}$, we compute the perturbation loss $\psi(\mathbf{w}, \mathbf{v})$, and iteratively solve for the worst-case direction $\mathbf{v}^*$ via power iteration on the Hessian of $\psi$, i.e., (Golub \& der Vorst, 2000) of our manuscript, ensuring maximal disruption to the inversion consistency. The worst-case direction $\mathbf{v}^*$ is then used to optimize our encoder to achieve the manipulation inversion. We further provide [Algorithm 1](https://anonymous.4open.science/r/icml2025_14926/alg.pdf) to depict the overall pipeline of our method.
>
> **[Q3 Experimental Designs Or Analyses: Editing Performance]**
>
> Indeed, for GAN inversion, the evaluations on editing performances are *ad hoc* due to the lack of ground-truth, as also pointed out by the reviewer. This also motivates us to establish a new proxy to evaluate the editing realism, i.e., the accuracy of manipulation inversion, with proved equivalence via Lemma 4.2 of our manuscript. Therefore, Fig. 4 of our manuscript demonstrates that for various editing directions, our method is able to precisely invert back, thus equally proving the superior editing performances of our method.
>
> We further conducted new evaluations based on the suggestions from the reviewer, i.e., using widely applied FID, ID, and Clip scores to evaluate the editing performance in face scenarios, together with the ClipDiff and ClipIQA scores for church scenarios. Please note that for face scenarios, we are aware of the semantics of editing attribute, thus capable of calculating Clip scores as suggested by the reviewer. For church scenarios, however, we cannot access the ground-truth semantics to edit attribute, thus infeasible to calculate IDs and Clip scores. Instead, we employ the editing directions by GANSpace and calculate the difference of Clip features between unedited and edited images (named as ClipDiff), to verify the effectiveness of editing. This is calculated by
>
> $$
> \mathrm{ClipDiff} = 1 - \mathrm{cos}\left(\phi(\mathbf{x}\_\mathrm{original}),\phi(\mathbf{x}_{\mathrm{edited}})\right)
> $$
>
> where $\phi(\cdot)$ denotes Clip image encoder. Moreover, since church scenarios only contain 300 test images, the FIDs can significantly vary, and we calculate ClipIQA scores instead to evaluate the subjective quality of edited images.
>
> We report the results in [Table 1](https://anonymous.4open.science/r/icml2025_14926/table_edit.pdf), in which our method consistently achieves the best editing performances, across different scenarios and attributes. Our superior editing performance is also in accordance with the best performance of manipulation inversion, in which their equivalence has been pointed out in our manuscript.
>
> **[Q4 Relation to Broader Scientific Literature]**
>
> Yes, using human feedback or reinforcement learning instead of our VAT mechanism is expected to be useful when extending our manipulation inversion method to large-scale generative models, in which the latent space becomes further complicated. We leave this as our interesting future work.

---

### Decision · Program_Chairs · 2025-05-01

**Decision:**

Reject

**Comment:**

The paper proposes a manipulation-aware GAN inversion method with solid analysis of StyleGAN’s latent space and some promising results on identity preservation. However, after reviewing the paper and the rebuttal, I’m recommending rejection in its current form. A key issue raised by multiple reviewers, and one I agree with, is the lack of proper evaluation on editing performance. Invertibility does not imply editability, and without strong evidence that the proposed method leads to better or more controllable edits, the main claim falls short.

As an area chair with experience in GAN and diffusion-based editing, I also find it limiting that the paper focuses solely on StyleGAN, without citing or comparing to more recent diffusion-based editing methods. These models are now dominant in the field, and ignoring them weakens the paper’s relevance. It would also be valuable to explore whether the method generalizes to diffusion latent spaces like H-space. I encourage the authors to revise the paper accordingly and submit to a future venue.